# Ezrin activation by LOK phosphorylation involves a PIP$_2$-dependent wedge mechanism

Thaher Pelaseyed[1,2], Raghuvir Viswanatha[1,2,3], Cécile Sauvanet[1,2], Joshua J Filter[2], Michael L Goldberg[2], Anthony Bretscher[1,2]*

[1]Weill Institute for Molecular and Cell Biology, Cornell University, Ithaca, United States; [2]Department of Molecular Biology and Genetics, Cornell University, Ithaca, United States; [3]Department of Genetics, Harvard Medical School, Boston, United States

**Abstract** How cells specify morphologically distinct plasma membrane domains is poorly understood. Prior work has shown that restriction of microvilli to the apical aspect of epithelial cells requires the localized activation of the membrane-F-actin linking protein ezrin. Using an *in vitro* system, we now define a multi-step process whereby the kinase LOK specifically phosphorylates ezrin to activate it. Binding of PIP$_2$ to ezrin induces a conformational change permitting the insertion of the LOK C-terminal domain to wedge apart the membrane and F-actin-binding domains of ezrin. The N-terminal LOK kinase domain can then access a site 40 residues distal from the consensus sequence that collectively direct phosphorylation of the appropriate threonine residue. We suggest that this elaborate mechanism ensures that ezrin is only phosphorylated at the plasma membrane, and with high specificity by the apically localized kinase LOK.

*For correspondence: apb5@cornell.edu

**Competing interests:** The authors declare that no competing interests exist.

## Introduction

All nucleated cells can polarize to generate morphologically and biochemically distinct regions at the cell surface. For example, the apical and basolateral domains of epithelial cells have distinct protein and lipid compositions, and microvilli are restricted to the apical domain. How cells maintain morphologically distinct regions of their cell surface is not clear.

We have been addressing this issue by examining how microvilli are assembled and specifically localized to the apical surface of epithelial cells (*Sauvanet et al., 2015*). A critical component of epithelial microvilli is ezrin, the founding member of the closely-related ezrin/radixin/moesin (ERM) protein family, that serves as a regulated membrane-F-actin linking protein (*Bretscher, 1983*, *1989*; *Fehon et al., 2010*). Genetic knockout of ezrin in the mouse results in enterocytes with shorter and disorganized microvilli. In the fruit fly, loss of the single ERM protein is lethal, but when selectively knocked out in photoreceptor cells, microvilli are lost (*Karagiosis and Ready, 2004*; *Saotome et al., 2004*; *Speck et al., 2003*). Thus, ERM proteins provide a critical function in polarized morphogenesis.

ERM proteins are regulated by a reversible head-to-tail interaction (*Figure 1A*). Like all ERMs, ezrin contains an N-terminal FERM domain that binds the plasma membrane and a C-terminal F-actin-binding domain (ezrin-CTD) that can attach to the underlying actin filaments that make up the core of microvilli (*Gould et al., 1989*; *Turunen et al., 1994*). In the closed inactive state, the FERM domain is tightly associated with the ~80 residues of ezrin-CTD, masking the membrane association and F-actin-binding sites (*Gary and Bretscher, 1995*; *Pearson et al., 2000*; *Reczek and Bretscher, 1998*). Linking these two regions is a ~150 residue α-helical region that folds into an anti-

**Figure 1.** *In vitro* phosphorylation of full-length ezrin requires PIP$_2$ and LOK C-terminal domain. (**A**) Left panel: A cartoon illustration of cytoplasmic closed/inactive ezrin versus membrane-tethered open/active ezrin acting as crosslinker between the plasma membrane (PM) and the cytoskeletal F-actin. Right panel: The domain structure of ezrin and LOK constructs used in this study. The numbers indicate amino acids residues at protein domain boundaries. (**B**) *In vitro* kinase assay showing that 10 nM LOK phosphorylates 18 µM ezrin-CTD. Data are presented as mean ± SE, n = 3, two-way

*Figure 1 continued on next page*

*Figure 1 continued*

ANOVA (See also *Figure 1—source data 1*), ****$p<0.0001$. (C) *In vitro* kinase assay showing LOK-mediated phosphorylation of full-length ezrin in presence of 90 µM of $PIP_2$ micelles. Data are presented as mean ± SE, n = 3, two-way ANOVA (See also *Figure 1—figure supplement 1* and *Figure 1—source data 1*), ****$p<0.0001$. (D) *In vitro* kinase assay showing that 18 µM ezrin is specifically primed by 90 µM of $PIP_2$ micelles and not by $IP_3$ or other phospholipids at 90 µM concentrations. Blots are derived from same membrane. Data are presented as mean ± SE, n = 3, two-way ANOVA (See also *Figure 1—source data 1*), ****$p<0.0001$. (E) *In vitro* kinase assay showing that unilamellar liposomes DOPC:$PIP_2$ (90 mol% DOPC, 10% $PIP_2$) or DOPC:DOPS:$PIP_2$ (80 mol% DOPC, 10 mol% DOPS, 10 mol% $PIP_2$) promote phosphorylation of 18 µM ezrin by 10 nM LOK, whereas DOPC (100 mol% DOPC) or DOPC:DOPS (70 mol% DOPC, 30 mol% DOPS) fail to promote LOK-mediated ezrin phosphorylation. (F–G) 10 nM LOK-N phosphorylates 18 µM ezrin-CTD but not full-length ezrin in presence of 90 µM $PIP_2$. Data are presented as mean ± SE, n = 3, two-way ANOVA (See also *Figure 1—source data 1*), ***$p<0.0002$, ****$p<0.0001$. Total ezrin is shown in red and phosphorylation of T567 in green in dual color Western blots.

The following source data and figure supplement are available for figure 1:

**Source data 1.** Experimental replicates for *Figure 1B,C, D, F and G*.

**Figure supplement 1.** Phosphorylation of full-length ezrin by full-length LOK requires $PIP_2$ micelles or $PIP_2$-containing liposomes.

parallel coiled coil hairpin in the closed structure (*Li et al., 2007*). In the open structure, the unmasked FERM domain binds the plasma membrane, the unmasked C-terminal domain binds F-actin, and the α-helical region presumably unravels. The unmasked FERM domain can bind many proteins, including the NHERF family adaptor proteins EBP50 and E3KARP, and the trans-membrane proteins CD44, ICAMs, and β-dystroglycan (*Mori et al., 2008*; *Reczek et al., 1997*).

The transition between closed and open ERMs requires phosphorylation of a specific threonine residue (T567 in ezrin; T564 in radixin and T558 in moesin) (*Nakamura et al., 1995*; *Simons et al., 1998*). Additionally, *in vivo* phosphorylation was found to require binding of the FERM domain to the plasma membrane phospholipid phosphatidylinositol 4,5-bisphosphate ($PIP_2$) (*Fievet et al., 2004*; *Hao et al., 2009*). Structural analysis of the radixin FERM domain in the presence or absence of $IP_3$ led to the suggestion that $PIP_2$ binding to FERM directly changes the conformation so as to partially release the C-terminal domain to make the phosphorylation site accessible (*Hamada et al., 2000*). Consistent with this model, in the closed configuration, the T567 that becomes phosphorylated is buried within the interface between the FERM and the ezrin-CTD (*Li et al., 2007*; *Pearson et al., 2000*).

Despite the findings above, how ezrin activation is restricted to the apical membrane was not resolved. At steady state, about one half of the ezrin in an epithelial cell is phosphorylated. Moreover, *in vivo* phosphorylation of ezrin turns over on the time-scale of minutes, similar to the lifetime of microvilli. These observations strongly suggest a morphogenetic principle: a dynamic system of local ezrin activation by phosphorylation coupled with delocalized deactivation by dephosphorylation. Lymphocyte-oriented-kinase (LOK) and its close paralog Ste20-like-kinase (SLK) were then found to be the major ezrin kinases in epithelial cells. Remarkably, LOK and SLK are currently the only kinases known to selectively localize to the apical membrane of epithelial cells (*Viswanatha et al., 2012*). LOK had previously been identified as the kinase that phosphorylates ezrin in lymphocytes (*Belkina et al., 2009*).

LOK and SLK belong to the germinal center-like kinase (GCK) -V subfamily of kinases. They consist of a conserved N-terminal kinase domain, a less conserved intermediate region, and a moderately conserved C-terminal domain (*Figure 1A*). A single ortholog exists in *Drosophila melanogaster* (Slik) and *Caenorhabditis elegans* (GCK4). In the fly, Slik has been identified as the sole kinase that phosphorylates the single ERM ortholog, moesin (*Carreno et al., 2008*; *Hipfner et al., 2004*; *Kunda et al., 2008*). It is notable that LOK is highly selective for members of the ERM family, as the kinase's target sequences requires a conserved tyrosine at position −2 relative to the substrate threonine (*Belkina et al., 2009*).

Here, we describe an *in vitro* system to reveal why ezrin has to bind $PIP_2$ to become a substrate for phosphorylation by LOK. As far as we are aware, this is the first example of a protein that has to bind a phosphoinositide lipid to serve as a kinase substrate. Our results indicate that $PIP_2$ binding to the ezrin FERM domain transmits a conformational change through the α-helical region to weaken the FERM/ezrin-CTD association. $PIP_2$-priming of ezrin permits the LOK C-terminal domain to act as

a wedge between the FERM and ezrin-CTD to allow the kinase domain to gain access and phosphorylate T567. In support of this model, we design a chimeric protein and show that it can recapitulate the activity of LOK *in vitro* and *ex vivo*. Further, we define a distal site in ezrin about 40 residues from T567 that is also necessary for phosphorylation by the kinase domain. In addition to being required for ezrin phosphorylation, the LOK C-terminal domain negatively regulates the kinase activity of LOK. Thus, the PIP2-dependent mechanism of ezrin phosphorylation by LOK involves several distinct steps, thereby ensuring the specificity of the reaction.

## Results

### *In vitro* phosphorylation of ezrin by LOK requires PIP2 and the LOK C-terminal domain

We first established an *in vitro* phosphorylation assay for purified LOK using the isolated ezrin-CTD as substrate. LOK phosphorylated the ezrin-CTD as detected using pT567 antibody (*Figure 1B*), but failed to phosphorylate full-length ezrin (*Figure 1C*). We therefore explored the possible role of phospholipid phosphatidylinositol 4,5-bisphosphate (PIP2) as previous reports have suggested a regulatory role for this phospholipid (*Fievet et al., 2004*; *Hao et al., 2009*). In the presence of PIP2 micelles, LOK was readily able to phosphorylate ezrin (*Figure 1C*), but PIP2 did not influence phosphorylation of the ezrin-CTD by LOK (*Figure 1—figure supplement 1A*). The requirement for PIP2 was specific, as other phosphoinositides, or soluble IP3, failed to substitute for PIP2 in phosphorylation of ezrin (*Figure 1D*). The requirement of PIP2 for ezrin phosphorylation was not simply a negative charge effect as PIP2-dependency was reproduced in 100-nm-sized synthetic unilamellar liposomes of DOPC or DOPC:DOPS containing 10 mol% PIP2, which represent a more physiological system (*Figure 1E*).

We next explored which domains of LOK are required for ezrin phosphorylation. LOK consists of an N-terminal kinase domain (LOK-N) and a C-terminal region (LOK-C) containing putative polo kinase kinase (PKK) domains of unknown function (*Figure 1A*). *In vitro*, both full-length LOK and LOK-N can phosphorylate ezrin-CTD, although the isolated kinase domain was less active (*Figure 1F*). Whereas intact LOK phosphorylated ezrin in a PIP2-dependent manner, LOK-N was unable to phosphorylate ezrin, even in the presence of PIP2 micelles and unilamellar DOPC:DOPS liposomes containing 10 mol% PIP2 (*Figure 1G* and *Figure 1—figure supplement 1B*). We conclude that *in vitro* phosphorylation of full-length ezrin requires both PIP2 and the C-terminal domain of LOK.

### *In vitro* phosphorylation requires active participation by the central α-helical domain of ezrin

The FERM domain of ezrin is linked to the ezrin-CTD by a ~150 residue α-helical coiled coil hairpin, as represented in the structure of the close homolog *Sf*moesin (*Figure 2A*) (*Li et al., 2007*). This region consists of three α-helices that connect the FERM domain to the ezrin-CTD via an antiparallel coiled coil (*Li et al., 2007*). In the absence of the α-helical region, the individual FERM and ezrin-CTD domains bind avidly together (*Pearson et al., 2000*). To explore whether binding of PIP2 to the FERM domain simply weakens the association with the ezrin-CTD as proposed (*Hamada et al., 2000*), we asked whether a complex of isolated GST-ezrin-CTD and FERM could be phosphorylated by LOK or LOK-N in the presence of PIP2. Remarkably, while GST-ezrin-CTD was phosphorylated by LOK and LOK-N, we found that the isolated GST-ezrin-CTD+FERM complex could not be phosphorylated by LOK or LOK-N in the presence of PIP2 (*Figure 2B*). Furthermore, LOK failed to phosphorylate GST-ezrin-CTD in complex with FERM in presence of DOPC:DOPS:PIP2 unilamellar liposomes (*Figure 2—figure supplement 1*). We conclude that the central α-helical region is required for translating PIP2-binding to the FERM domain into a weakened association between FERM and ezrin-CTD domains, thus allowing *in vitro* phosphorylation of ezrin by LOK.

### PIP2 primes ezrin to serve as an efficient substrate for phosphorylation by LOK

Having established the broad features of the PIP2-dependent phosphorylation of ezrin by LOK, we explored the kinetic parameters of the phosphorylation reactions in order to gain more insight into

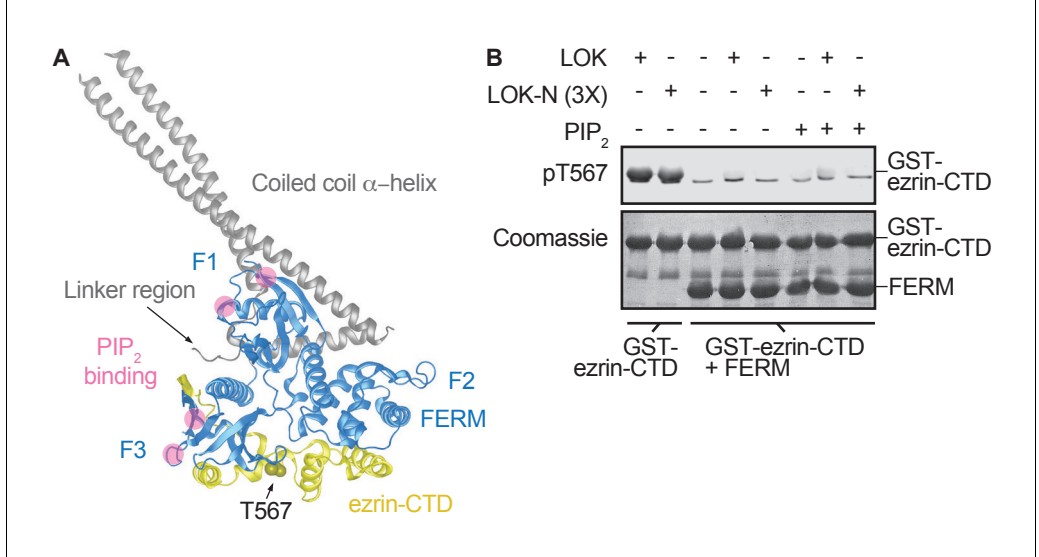

**Figure 2.** The central α-helical coiled coil hairpin region of ezrin is required for PIP$_2$-dependent phosphorylation. (**A**) Crystal structure of the ERM protein *Sf*moesin (PDB: 2I1K) in a closed/inactive state. Lobes F1-3 represent the lobes in FERM domain (blue). The α-helical coiled coil hairpin is shown in gray. An acidic linker region (gray) of the hairpin occupies a PIP$_2$-binding site (pink), embedded between F1 and F3 lobes. T567 represents the position of ezrin T567 superimposed on *Sf*moesin-CTD (yellow). This residue is masked in this closed state. (**B**) 10 nM LOK and 10 nM LOK-N phosphorylate 18 μM GST-ezrin-CTD, but both LOK and LOK-N fail to phosphorylate 18 μM of a GST-ezrin-CTD+FERM complex in absence or presence of 90 μM PIP$_2$. Phosphorylation of GST-ezrin-CTD was detected with pT567 antibody. Total protein was visualized using Coomassie.

The following figure supplement is available for figure 2:

**Figure supplement 1.** GST-ezrin-CTD+FERM complex, lacking the central α-helix region, is not phosphorylated in presence of PIP$_2$ micelles or PIP$_2$-containing liposomes.

the mechanism (*Table 1* and *Figure 1—figure supplement 1*). Full-length LOK and just the kinase domain, LOK-N, had about the same affinity for the ezrin-CTD ($K_m$ 252 and 176 μM, respectively), but the maximal turnover rate for LOK-N was lower ($k_{cat}$ of 19.8 min$^{-1}$ for LOK versus 1.4 min$^{-1}$ for LOK-N). In the presence of PIP$_2$, LOK phosphorylated ezrin with a $K_m$ of 5 μM and displayed a lower maximal turnover ($k_{cat}$ of 8.5 min$^{-1}$). In the absence of PIP$_2$, LOK hardly phosphorylated ezrin, and the $K_m$ was difficult to measure, but was of the order of 800 μM. Thus, PIP$_2$ enhances the affinity of

**Table 1.** A comparison of the kinetic constants for kinase-substrate pairs based on [32]P incorporation. Concentration of kinase in each experiment was 10 nM. Data are represented as mean ± SE, n = 3 (See also *Figure 1—figure supplement 1C–D* and *Table 1—source data 1*).

| Kinase/Substrate | $K_m$[a] (μM) | $V_{max}$ (pmole×min$^{-1}$) | $k_{cat}$[a] (min$^{-1}$) | $k_{cat}/K_m$ (μM$^{-1}$×min$^{-1}$) |
|---|---|---|---|---|
| LOK/ezrin + PIP$_2$ | 5.5 ± 1.6 | 0.85 ± 0.07 | 8.5 ± 0.7 | 1.56 |
| LOK/ezrin | 782 ± 197 | 0.56 ± 0.10 | 5.6 ± 1.0 | 0.01 |
| LOK/ezrin-CTD | 252 ± 27 | 1.98 ± 0.16 | 19.8 ± 1.4 | 0.08 |
| LOK-N/ezrin-CTD | 176 ± 30 | 0.14 ± 0.01 | 1.4 ± 0.1 | 0.01 |

[a]$K_m$ and $k_{cat}$ were calculated using equations 1 and 2.
Source data 1. Data summary and analysis for *Table 1*, based on *Figure 1—figure supplement 1C–D*.

LOK for ezrin by about 160 fold. LOK-N phosphorylated intact ezrin so poorly that it was not possible to derive a $K_m$ value, emphasizing the importance of LOK C-terminal domain. Our data reveal that the catalytic efficiency ($k_{cat}/K_m$) of LOK for ezrin and ezrin-CTD is 1.56 and 0.08 $\mu M^{-1}min^{-1}$, respectively. Thus, as a substrate for LOK, ezrin is about twenty-fold better than the ezrin-CTD. Since the affinity of LOK for ezrin is greatly enhanced by PIP$_2$, and this affinity is much higher than LOK for the ezrin-CTD, the LOK C-terminal domain (LOK-C) must actively engage full-length ezrin in a PIP$_2$-dependent manner.

## The LOK C-terminal domain serves as a wedge between the FERM and ezrin-CTD domains

To explore if LOK-C binds specific domains of ezrin, equilibrium-binding curves were generated. We measured the depletion of LOK-C from the supernatant of reactions containing sedimented beads with either immobilized ezrin-CTD or FERM domain (*Figure 3A and B*). Measurements revealed that the LOK-C binds both the free FERM and ezrin-CTD domains with a $K_d$ of about 10 µM and 2 µM, respectively. We also attempted to measure the association of the LOK-C with full-length ezrin in the absence and presence of PIP$_2$, but minimal binding could be detected (data not shown). Thus, priming of ezrin by PIP$_2$ does not unmask binding sites sufficiently to enable the determination of a binding constant.

The combination of the kinetics and binding experiments lead to a model for the basic mechanism that allows the LOK kinase domain to gain access to T567 (*Figure 3C*). Ezrin binds PIP$_2$ and this induces a conformational change that is transmitted through the α-helical linker domain to transiently loosen the association between the FERM and ezrin-CTD. The loosening of FERM-ezrin-CTD interaction allows LOK-C to enter between the FERM and ezrin-CTD - working like a wedge - to render T567 accessible to the kinase domain.

## The PIP$_2$-dependent wedge mechanism of ezrin phosphorylation can be recapitulated with a chimeric kinase

Our model implies that site(s) in ezrin for binding LOK-C are masked in the closed, inactive state. This is also true of the binding site for ezrin-binding protein of 50 kD (EBP50). EBP50 consists of two PDZ domains and a C-terminal tail region. The latter binds tightly to a surface on the FERM domain that is masked in closed full-length ezrin (*Reczek et al., 1997*; *Reczek and Bretscher, 1998*). Structural work has shown that the C-terminal tail of EBP50 (EBP50t) binds to the same surface of the FERM domain that is occupied by a C-terminal α-helix belonging to ezrin-CTD, thereby explaining why the binding site for EBP50t is masked in the full-length inactive protein (*Finnerty et al., 2004*). We therefore tested if EBP50t could compensate functionally for the LOK C-terminal domain in phosphorylation of PIP$_2$-primed ezrin. We generated and purified a LOK-N-EBP50t chimera (*Figure 4A*) and used it in *in vitro* phosphorylation assays (*Figure 4B*). As expected, both LOK-N-EBP50t and LOK could phosphorylate the ezrin-CTD. When presented with full-length ezrin, neither kinase could phosphorylate ezrin in the absence of PIP$_2$, while both LOK and LOK-N-EBP50t could phosphorylate ezrin robustly when PIP$_2$ was included. Thus, the chimeric protein supports the PIP$_2$-dependent wedge model, by interaction of C-terminal EBP50t with ezrin FERM, and consequently compensating for LOK-C, when ezrin is primed by PIP$_2$.

To see if LOK-N-EBP50t could mimic LOK *ex vivo*, we explored its localization in cultured placental choriocarinoma JEG3 cells in which LOK has been shown to selectively localize to the apical domain (*Viswanatha et al., 2012*). Upon transient expression, both LOK and LOK-N-EBP50t localized to microvilli, whereas LOK-N did not (*Figure 4C*). To see if LOK-N-EBP50t could compensate functionally for LOK or SLK, using CRISPR/Cas9, we generated a JEG-3 cell line lacking LOK and knocked down endogenous SLK by siRNA expression. As with double siRNA knockdown (*Viswanatha et al., 2012*), kinase suppression reduced ezrin T567 phosphorylation by over 80% and resulted in loss of microvilli (*Figure 4—figure supplement 1A and B*). Introduction of LOK or LOK-N-EBP50t into these cells restored microvilli, whereas LOK-N failed to restore microvilli (*Figure 4D*). LOK-N-EBP50t also restored ezrin phosphorylation in kinase-suppressed cells (*Figure 4—figure supplement 1C*). Thus, LOK-N-EBP50t is able to recapitulate both the enzymatic properties of LOK as well as its function in microvilli formation, via restricting activated ezrin to the apical membrane (*Figure 4E*).

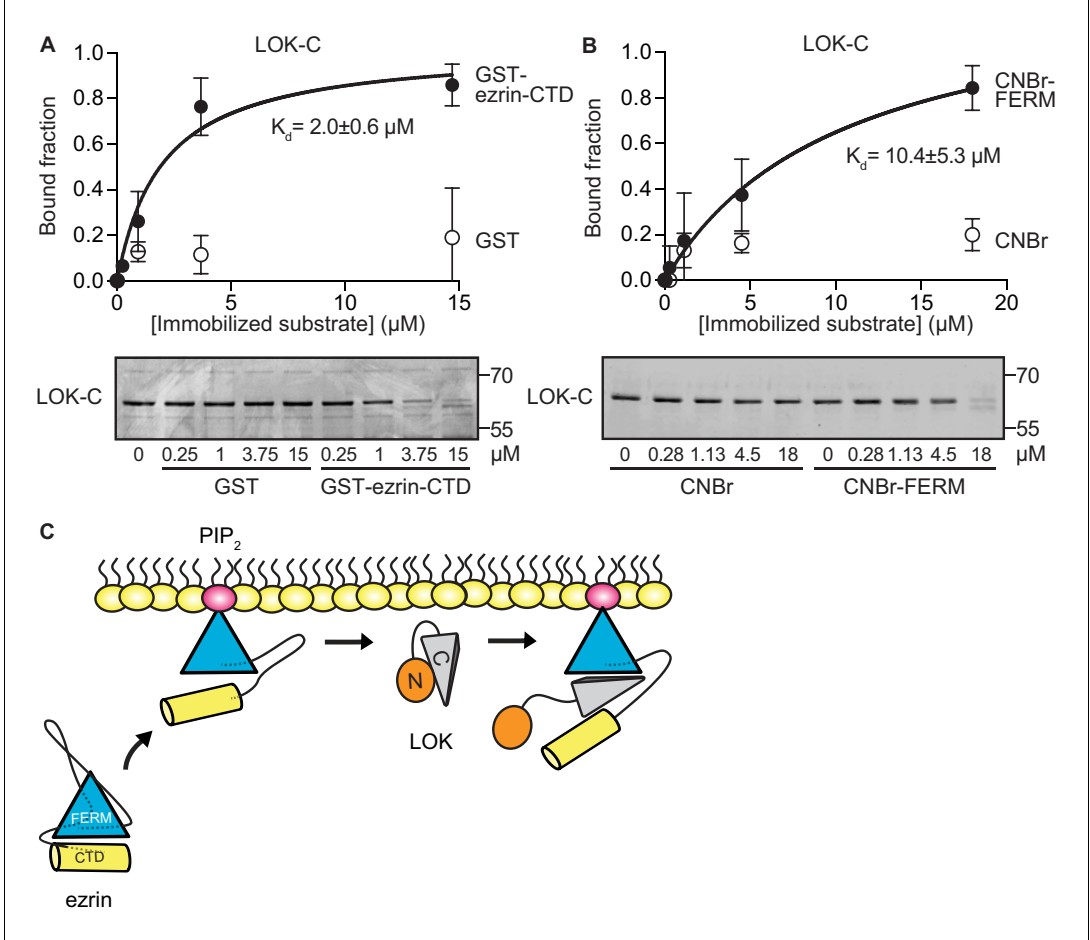

**Figure 3.** LOK C-terminal domain binds both ezrin FERM and C-terminal domains. (**A**) A binding curve for binding of 0–15 μM immobilized GST-ezrin-CTD to 200 nM LOK-C at $K_d = 2.0 \pm 0.6$ μM. Bottom panel depicts representative experiment and Coomassie staining of unbound fraction of LOK-C. Data are represented as mean ± SE, n = 3 (See also *Figure 3—source data 1*). (**B**) Binding curve for binding of 0–18 μM immobilized CNBr-FERM to 200 nM LOK-C at $K_d = 10.4 \pm 5.3$ μM. Bottom panel depicts representative experiment and Coomassie staining of unbound fraction of LOK-C. Data are represented as mean ± SE, n = 3 (See also *Figure 3—source data 1*). (**C**) A model of LOK-C acting as a wedge that pries open FERM and ezrin-CTD in a PIP₂-primed ezrin molecule.

The following source data is available for figure 3:

**Source data 1.** Analysis for binding curves in *Figure 3A–B*.

## The LOK C-terminal domain is involved in auto-inhibition

High expression of LOK-C in JEG-3 cells inhibits ezrin phosphorylation and results in the loss of microvilli. However, in low-level expressing cells, LOK-C is localized to regions of individual microvilli with local loss of ezrin (*Viswanatha et al., 2012*). Thus in cells, LOK-C has the properties of a negative regulatory domain.

We therefore examined the effect of LOK-C in the phosphorylation of ezrin or ezrin-CTD *in vitro*. In these assays, we employed LOK or LOK-N-EBP50t at 10 nM, or LOK-N at 30 nM and the ezrin or ezrin-CTD substrates at 18 μM. In the presence of 90 μM PIP₂, LOK-C inhibited the ability of LOK or LOK-N-EBP50t to phosphorylate ezrin with an IC₅₀ of 1.9 ± 0.1 μM and 1.6 ± 0.1 μM, respectively (*Figure 5A and C*). Similarly, the inhibition curves with ezrin-CTD as substrate showed an IC₅₀ for LOK of 1.9 ± 0.2 μM and for LOK-N of 1.2 ± 0.2 μM (*Figure 5B and C*). As these IC₅₀ values were all much lower than the 18 μM substrate concentration, inhibition by LOK-C is due to a direct effect on the LOK kinase activity rather than affecting the substrate. These similar IC₅₀ values also suggest that the mechanism of inhibition is similar in all cases, namely competitive inhibition. Moreover, since

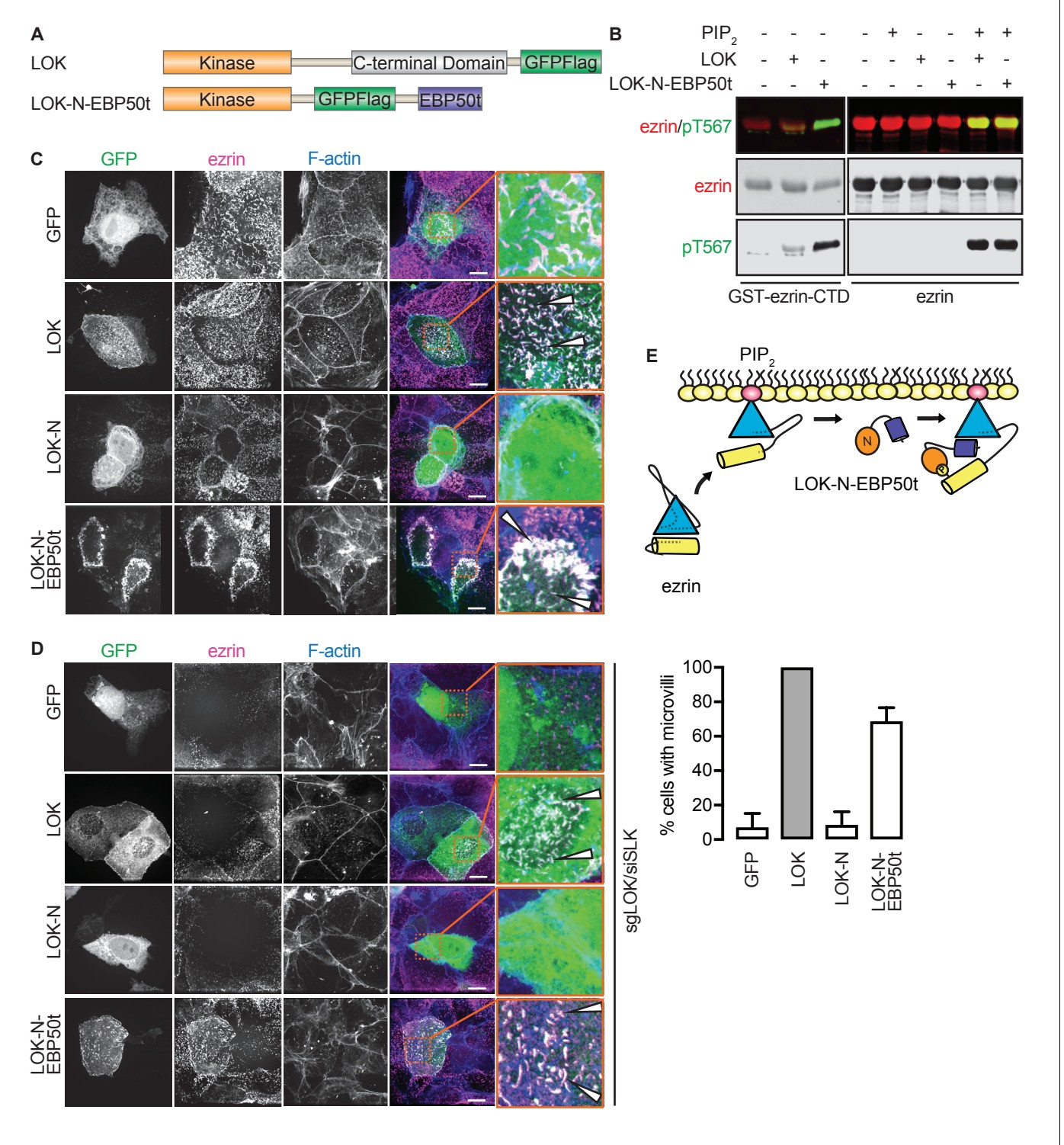

**Figure 4.** The tail of EBP50 compensates for lack of LOK C-terminus in $PIP_2$-dependent ezrin phosphorylation. (**A**) A schematic protein domain comparison of full-length LOK fused to GFP-Flag tag (green) and LOK-N fused to EBP50 tail (purple) via an internal GFP-Flag tag. (**B**) *In vitro* kinase assay showing that 10 nM LOK and 10 nM LOK-N-EBP50t both phosphorylate 18 µM ezrin-CTD, but only phosphorylate 18 µM ezrin in presence of 90 µM $PIP_2$. Total ezrin is shown in red and phosphorylation of T567 in green in dual color western blots. (**C**) Maximum projections of wild type JEG-3 cells overexpressing GFP, and GFP fusions of LOK, LOK-N or LOK-N-EBP50t (green) show that LOK and LOK-N-EBP50t are targeted to microvilli, while GFP and LOK-N are cytoplasmic. Ezrin in magenta. F-actin in blue. Insets represent area within dotted boxes. Arrows indicate microvilli. Scale bars, 10 µm. (**D**) Representative maximum projections of ERM kinase-suppressed sgLOK/siSLK JEG-3 cells overexpressing GFP, and GFP fusions of LOK, LOK-N or

*Figure 4 continued on next page*

*Figure 4 continued*

LOK-N-EBP50t (green) show that LOK and LOK-N-EBP50t rescue microvilli, while GFP and LOK-N failed to rescue microvilli. Ezrin in magenta. F-actin in blue. Insets represent area within dotted boxes. Arrows indicate microvilli. Scale bars, 10 µm. Bar graph shows quantification of confocal images. Data are represented as mean ± SE, n = 3. (See also *Figure 4—source data 1*) (E) A model depicting how LOK-N-EBP50t binding to FERM in $PIP_2$-primed ezrin resembles the wedging of ezrin by LOK-C prior to phosphorylation of T567.

The following source data and figure supplement are available for figure 4:

**Source data 1.** Source data for quantification of microvilli in *Figure 4D* and *Figure 4—figure supplement 1B*.

**Figure supplement 1.** LOK-N-EBP50t rescues ezrin phosphorylation in kinase-suppressed JEG-3 cells.

the $IC_{50}$ values of LOK-C for LOK and LOK-N phosphorylation of the ezrin-CTD are similar, binding of the kinase domain of full-length LOK to the ezrin-CTD likely relieves the inhibition imposed by its own C-terminal domain. We therefore explored whether regions in the ezrin-CTD outside the minimal consensus sequence surrounding T567 might bind LOK-N and potentially relieve the inhibition imposed by LOK-C.

## LOK-N requires a docking site distal to the T567 consensus sequence

Kinases require a well-defined consensus sequence immediately flanking the phosphorylation site. In the case of LOK, there is a strong preference for a tyrosine at P-2 (Y565), which is conserved in all ERM proteins (*Belkina et al., 2009*). In order to explore the presence of additional regions that contribute to specific LOK phosphorylation of ezrin, we used N-terminal truncations of the ezrin-CTD to assess their ability to serve as substrates for LOK and LOK-N. As shown earlier, LOK-N has a slightly lower activity than LOK for GST-ezrin-CTD (*Figure 6A*). Robust phosphorylation was seen up to GST-ezrin-520–585. The shorter constructs, GST-ezrin-530–585 and GST-ezrin-555–585, were very poor substrates for LOK and LOK-N. As this suggests that the region 520–530 is important for recognition by LOK-N, we made a construct, GST-ezrin-490-(Δ520–530)−585, lacking this region. This truncated substrate was a significantly poorer substrate than GST-ezrin-490–585. Thus, the ezrin-CTD has an additional distal docking site approximately 40 residues N-terminal to the kinase consensus sequence.

## Discussion

In order to maintain the morphological polarity of a cell, specific morphogenetic components have to be controlled locally. In the case of microvilli on epithelial cells, ezrin is locally activated through T567 phosphorylation mediated by the closely related LOK and SLK kinases. Here, we define a novel multi-step mechanism for specific phosphorylation of ezrin by LOK (*Figure 6B*). In the first step, binding to $PIP_2$ induces a conformational change in ezrin. In the second step - that has to occur coincidentally with the first - the C-terminal domain of LOK (LOK-C) binds as a wedge between the FERM and ezrin-CTD domains. The wedge mechanism allows for the third step in which the kinase domain binds to a distal docking site in the ezrin-CTD upstream of the T567 phosphorylation site. The final step can now proceed, namely the phosphorylation of T567 within the appropriate consensus sequence setting. LOK itself is negatively regulated by LOK-C, and binding ezrin appears to relieve this inhibition. In the following narrative, we expand on each of these steps.

The evidence for the first step is that LOK requires $PIP_2$ to phosphorylate ezrin (*Figure 6B*, step 1). Since $PIP_2$ is a regulatory lipid largely confined to the plasma membrane, the highly specific priming of ezrin by $PIP_2$ ensures that ezrin phosphorylation is restricted to this membrane compartment. Our *in vitro* studies show that both phosphatidylinositol 4,5-bisphosphate ($PIP_2$) micelles and $PIP_2$-containing unilamellar liposomes prime ezrin for LOK-mediated phosphorylation, while other phospholipids and more importantly, $IP_3$, the head group in direct physical contact with ezrin, fail to prime ezrin for phosphorylation. From these data, we conclude that priming of ezrin for phosphorylation requires an appropriate charge distribution provided by phosphate groups in the head group of the phospholipid phosphatidylinositol 4,5-bisphosphate, and a high local concentration of the head group that can only be achieved *in vitro* using micelles or liposomes. Earlier studies have

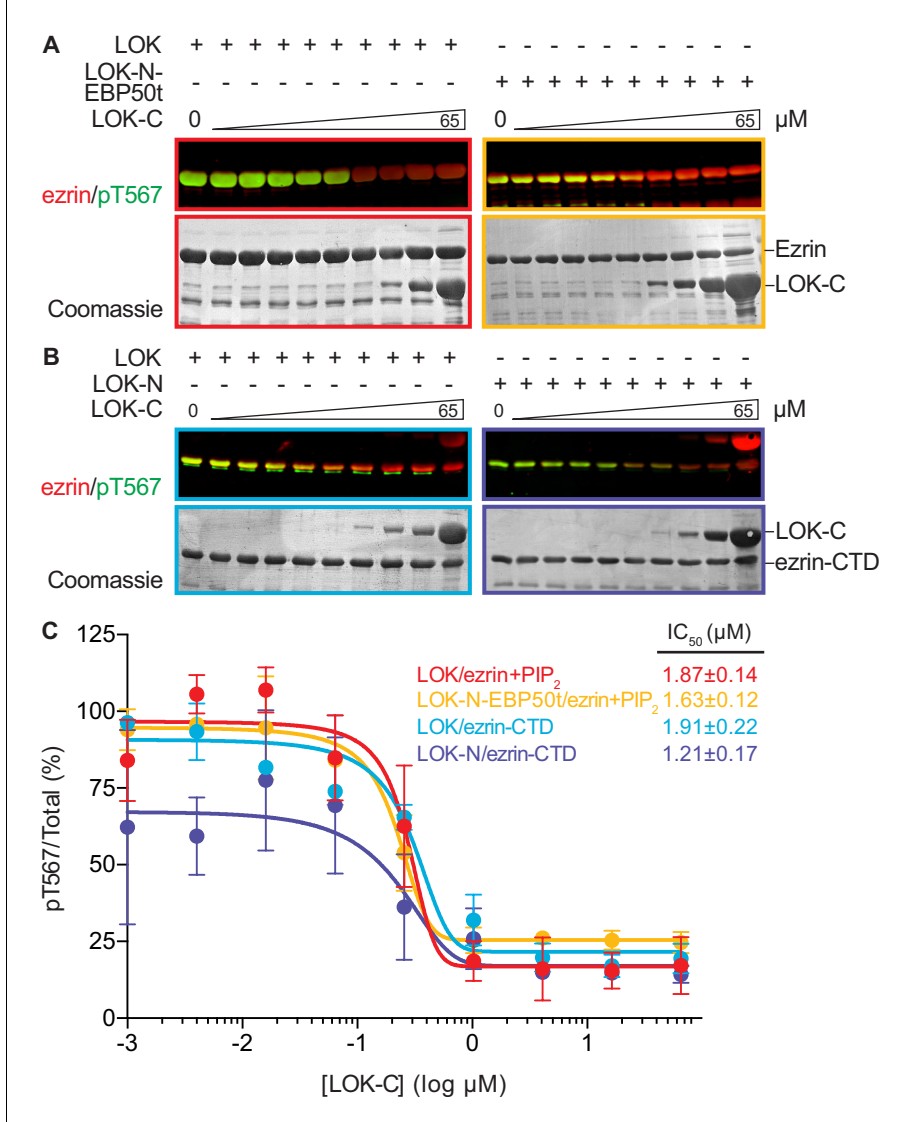

**Figure 5.** LOK C-terminus is a regulatory domain that inhibits the LOK kinase domain. (**A**) *In vitro* kinase assay showing that LOK-C (0–65 µM) inhibits phosphorylation of 18 µM ezrin phosphorylation by 10 nM LOK (red panel) and 10 nM LOK-N-EBP50t (yellow panel) in presence of 90 µM $PIP_2$. (**B**) *In vitro* kinase assay showing that LOK-C (0–65 µM) inhibits LOK-mediated phosphorylation of 18 µM ezrin-CTD (blue panel), and LOK-N-mediated phosphorylation of 18 µM ezrin-CTD (purple panel). Total ezrin is shown in red and phosphorylation of T567 in green in dual color western blots. (**C**) Inhibition curves of LOK/ezrin+$PIP_2$ (red), LOK-N-EBP50t/ezrin+$PIP_2$ (yellow), LOK/ezrin-CTD (blue), and LOK-N/ezrin-CTD (purple), with their corresponding $IC_{50}$-values. Data are represented as mean ± SE, n = 2 (see also *Figure 5—source data 1*).

The following source data is available for figure 5:

**Source data 1.** Data summary and analysis for *Figure 5C*, based on experimental replicates represented in *Figure 5A–B*.

shown that $PIP_2$ binds the FERM domain and it has been suggested that this interaction results in a conformational change to loosen the associated FERM-ezrin-CTD (**Ben-Aissa et al., 2012**; **Hamada et al., 2000**). Therefore, we were surprised to find that in the presence of $PIP_2$, LOK is unable to phosphorylate a complex of FERM associated with the ezrin-CTD. This revealed two points: First, $PIP_2$ acts through ezrin activation, and not by LOK activation, and second, the central

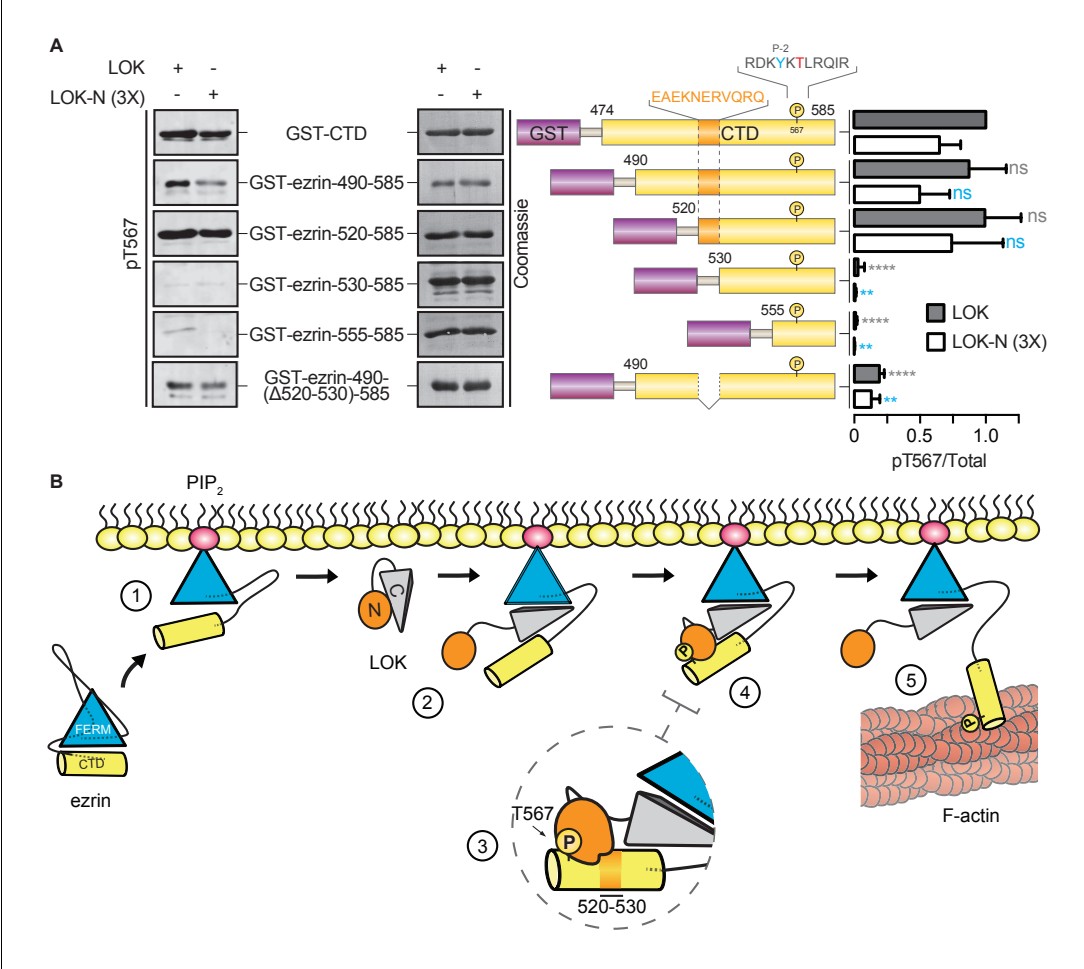

**Figure 6.** LOK kinase domain recognizes a docking site distal to T567 phosphorylation site. (**A**) *In vitro* kinase assay showing that 10 nM LOK and 30 nM LOK-N phosphorylate GST-ezrin-CTD, and the truncations GST-ezrin-490–585 and GST-ezrin-520–585. Phosphorylation decreases in the truncations GST-ezrin-530–585, GST-ezrin-555–585 and GST-ezrin-490(Δ520–530)−585. All substrate concentrations were 18 μM. Schematic illustration of GST-ezrin-CTD and truncations shows the distal docking site in orange letters and box. LOK kinase consensus sequence is shown in black, in which the T567 phosphorylation site is in red letters and the conserved Y565 is in blue letters. Data are represented as mean ± SE, n = 3, two-way ANOVA (See also *Figure 6—source data 1*), **p<0.0021, ****p<0.0001. (**B**) A working model depicting the proposed coincidence detection mechanism for ezrin phosphorylation. (1) Ezrin is recruited to PIP$_2$ and primed for phosphorylation. (2) Autoinhibition of LOK is relieved as LOK-C wedges in between FERM and ezrin-CTD. (3) LOK-N binds to distal docking site on ezrin-CTD and, (4) phosphorylates T567. (5) LOK remains tethered to activated ezrin, allowing phosphorylation of microvillar ezrin in a positive feedback loop.

The following source data is available for figure 6:

**Source data 1.** Experimental replicates for *Figure 6A*.

α-helical coiled coil hairpin connecting the FERM to the ezrin-CTD is necessary for priming of intact ezrin for phosphorylation. The detailed function of the central coiled coil hairpin is unknown, but one attractive possibility is that the α-helical coiled coil region acts as a spring counteracting the strong association between the FERM and ezrin-CTD, and upon binding PIP$_2$, the strength of the spring is enhanced. The proposed spring model has similarities in other proteins. In the ERM-related protein merlin, the central α-helical hairpin undergoes an unfolding event, like a 'switchblade', during opening of the protein (*Li et al., 2007*). Another example occurs within the influenza virus hemagglutinin protein. This protein is a membrane-fusion glycoprotein with a rod-shaped α-helical ectodomain that undergoes a dramatic 'switchblade-like' unfolding event at low pH (*Skehel and Wiley, 2000*).

The active involvement of LOK-C (*Figure 6B*, step 2) is shown by the finding that the isolated LOK kinase domain cannot phosphorylate ezrin under any condition, whereas it can phosphorylate the isolated ezrin-CTD. Moreover, in the presence of PIP$_2$, the $K_m$ of full-length LOK for ezrin is about 45 times lower than for the ezrin-CTD, indicating that the combined action of PIP$_2$ and the FERM domain greatly enhances specificity. Consistent with the kinetics of PIP$_2$-primed full-length ezrin, we found that LOK-C can bind with low affinity to both the isolated FERM and ezrin-CTD. Thus, PIP$_2$ priming of ezrin loosens the association between the FERM and ezrin-CTD, allowing LOK-C to pry the domains apart, acting like a wedge, and thus permitting the kinase domain access to T567 in the ezrin-CTD. Indeed, using a cross-linking approach *ex vivo*, we have shown that LOK is recovered more efficiently with open ezrin rather than with closed ezrin-T567A that cannot be phosphorylated, suggesting that LOK has an affinity for sites masked in closed ezrin (*Figure 6B*, step 5) (*Viswanatha et al., 2013*). Moreover, in support of this model, we have shown that a chimeric protein consisting of the LOK kinase domain fused to the domain of EBP50 that binds to a site on the FERM domain normally masked in the closed protein, can recapitulate the PIP$_2$-dependent phosphorylation of ezrin *in vitro* and also partially *ex vivo*.

In this study we reveal that LOK-C is a negative regulator of the kinase domain as it inhibits the activity of both LOK and LOK kinase domains toward ezrin or ezrin-CTD. Since LOK phosphorylates ezrin-CTD better than the isolated kinase domain does, binding the ezrin-CTD likely relieves the inhibition of full-length LOK by LOK-C in *cis*. LOK-C might have additional functional implications for kinase activity. The C-terminal domain of SLK shares 56% sequence identity with LOK-C and undergoes homodimerization, a process that enhances kinase activity *in vitro* and *ex vivo* (*Delarosa et al., 2011*; *Sabourin and Rudnicki, 1999*). Potential dimerization and autoactivation of LOK remains to be studied.

The specificity of kinases depends on local peptide consensus sequences, which in the case of ezrin phosphorylation by LOK involves the hydrophobic residue Y565 positioned -2 residues from T567 (*Belkina et al., 2009*). As is seen with some other serine/threonine kinases (*Biondi and Nebreda, 2003*; *Sharrocks et al., 2000*; *Ubersax and Ferrell, 2007*), we have also identified a distal docking site within residues 520–530 in the ezrin-CTD, required for the efficient phosphorylation of T567 by either LOK or the isolated kinase domain (*Figure 6B*, steps 3–4), thereby adding another level of specificity to the kinase reaction. Binding of the kinase to this site may also relieve the inhibition imposed by LOK-C in the context of the full-length LOK. It is interesting to note that the residues equivalent to ezrin 520–530 are relatively conserved, and Y565 and T567 perfectly conserved, in all ERM proteins, but not in the closely related merlin that also undergoes a similar FERM/merlin-CTD interaction, where the threonine equivalent to ezrin-T567 is not phosphorylated (*Nguyen et al., 2001*; *Sher et al., 2012*).

To maintain morphological polarity, there are multiple levels of selectivity that ensure the specific phosphorylation of ezrin by LOK. At the genetic level, cells lacking LOK exhibit greatly reduced ezrin phosphorylation (*Figure 4—figure supplement 1A*). On the subcellular level, LOK-C targets the kinase to the apical membrane (*Viswanatha et al., 2012*). At the molecular level, we have uncovered the simultaneous requirement for PIP$_2$ and LOK in an elaborate multiple-step mechanism to ensure specific and localized phosphorylation of ezrin. While substrate specificity of protein kinases has been the subject of many studies (*Ubersax and Ferrell, 2007*), fewer cases of conditional specificity have been examined in detail, with LOK-mediated ezrin phosphorylation being the first example of a specific requirement for a phosphoinositide binding to the substrate. As complex cell behaviors rely on the proper timing of phosphorylation events, such intimate co-regulation between substrates and kinases as described here may emerge as a general theme.

## Materials and methods

### Reagents

Ins(1,4,5)P$_3$ (catalog# 502), as well as PI (catalog# 003), PI(3)P (catalog# 910), PI(4)P (catalog# 912), PI(4,5)P$_2$ (catalog# 902), and PIP$_3$ (catalog# 908) in their respective acidic forms with di-palmitoyl (diC16) fatty acyl side chains, were purchased from Cell Signals, Inc (Colombus, OH). Adenosine 5′-triphosphate (ATP) disodium salt hydrate was obtained from Sigma-Aldrich (Saint Louis, MO). Ezrin mouse monoclonal antibody (DSHB Cat# CPTC-Ezrin-1 RRID:AB_2100318) was used at a dilution of

1:5000 for western blot and a rabbit polyclonal antibody raised against full-length human ezrin used at a dilution of 1:150 for immunofluorescence (*Bretscher, 1989*). Phospho-ezrin was detected using rabbit anti-pT567 antibody, raised against recombinant phosphopeptide CRDKYK(pT)LRQIR (*Hanono et al., 2006*; *Matsui et al., 1999*) at a dilution of 1:1000 for western blot. GFP monoclonal mouse antibody (Santa Cruz Biotechnology Cat# sc-9996 RRID:AB_627695) was used at a dilution of 1:1000 for western blot. LOK (Bethyl Cat# A300-399A RRID:AB_386110) and SLK (Bethyl Cat# A300-499A RRID:AB_451034) antibodies were used at 1:1000 and 1:500 dilutions, respectively, for western blot. F-actin was stained with Alexa Fluor 568-conjugated phalloidin (Thermo Fisher Scientific Inc., Waltham, MA) at a dilution of 1:150 in immunofluorescence.

## Plasmids and siRNAs

Human ezrin and ezrin FERM cDNAs have been described previously (*Chambers and Bretscher, 2005*; *Reczek et al., 1997*). Human LOK and LOK kinase domain (LOK-N) have been described previously (*Viswanatha et al., 2012*). GST-ezrin-CTD (474-585), ezrin-530–585, and ezrin-555–585 have been described previously (*Gary and Bretscher, 1995*). GST-ezrin-490–585 and ezrin-520–585 were cloned into the pGEX-3X plasmid (GE Healthcare Life Sciences, Pittsburg, PA) using BamHI and EcoRI restriction sites, using primers: ezrin-490–585 forward 5′-CGGTACGGATCCATGAGGGCGCA-GAGCCCACG-3, ezrin-520–585 forward 5′-CGGTACGGATCCAGGCAGAGAAGAACGAGCG-3′ and reverse 5′-GTACCGGAATTCTTACAGGGCCTCGAACTCG-3′. GST-ezrin-490-(Δ520–530)−585 was generated by phosphorylating primer 5′-GAGAAGCGCATCACTCTGCTGACGCTGAGC-3′ followed by an one-round assembly PCR. In detail, 10 pmoles primer was phosphorylated with T4 Polynucleo-tide kinase for 1 hr at 37°C. An one-round assembly reaction was set up containing 50 ng pGEX-3X-ezrin-490–585, 1 pmole phosphorylated primer, 10 mM dNTPs, 16 units Taq Ligase, 1× Taq Ligase Buffer and 0.4 units Phusion High-Fidelity DNA Polymerase. All reagents were obtained from New England BioLabs Inc.. The following PCR parameters were used: 95°C for 5 min; 30× (95°C for 1 min, 60°C for 1 min, 65°C for 7 min); 65°C for 7 min; hold at 12°C. The reaction was digested with DpnI prior to transformation into DH5α chemically competent cells (Thermo Fisher Scientific Inc., Waltham, MA). The chimeric protein LOK-N-GFPFlag-EBP50t was generated by amplifying LOK-N-GFP-Flag with primers: forward 5′- TTCACTGCGGCCGCATGGCTTTTGCCAATTTCCGCCGCATC-3′ and reverse 5′-CTTGTCATCGTCGTCCTTGTAGTCC-3′, followed by amplifying EBP50t (residues 321–358) using primers: forward 5′-GACGACGATGACAAGCTAGACTTCAACATCTCCCT-3′ and reverse 5′-CGCGCCGATCGTTAGAGGTTGCTGAAGAGTTCG-3′. The amplicons were fused using overlap-ping extension PCR and cloned into pQCXIP plasmid (Clontech Laboratories, Inc., Mountain View, CA) using NotI and PvuI restriction sites. pESUMO-LOK-C was generated by amplifying LOK resi-dues 499–968 using primers: forward 5′-GCCGTCTCAAGGTGGTACCAATCTCTCC-3′ and reverse 5′-CGTCTAGATTAAGAAGCATCCGCAGAACTGTA-3′ and cloning into pESUMO plasmid (LifeSen-sors Inc., Malvern, PA) using XbaI and BsmBI restriction sites. siRNA targeting SLK was Validated Silencer Select from Ambion with the following sequence: 5′-GCAGAAACAGACUAUCGAAdTdT-3′, as previously described (*Viswanatha et al., 2012*). siGL2, targeting GL2 luciferase 5′-CGUACGCG-GAAUACUUCGAdTdT-3′ has been described previously (*Hanono et al., 2006*).

## Protein expression and purification

GST and 6×His-SUMO fusion proteins were expressed in Rosetta 2(DE3)pLysS (EMD Millipore, Biller-ica, MA). In detail, cells were expanded in Terrific broth with antibiotics at 37°C up to OD 1.0. Pro-teins were expressed at 28°C for 4 hr by adding 1 mM IPTG. GST fusions were lysed in Lysis buffer (20 mM Tris pH 7.4, 140 mM NaCl, 1 mM EGTA, 1 mM DTT, 0.1% Triton X-100), cleared and filtered prior to binding to Glutathione-agarose (Sigma-Aldrich, Saint Louis, MO) (5 mg fusion protein per mL resin). Upon extensive wash in Lysis buffer, fusion proteins were eluted in Elution buffer (20 mM Tris pH 7.4, 140 mM NaCl, 1 mM EGTA, 1 mM DTT, 10 mM Glutathione). SUMO fusion protein was lysed in Binding buffer (20 mM sodium phosphate, 300 mM NaCl, 20 mM Imidazole, pH 7.4), cleared and filtered prior to binding to Ni-NTA agarose (Qiagen, Hilden, Germany) for 2 hr. SUMO fusion protein was eluted with Elution buffer (20 mM sodium phosphate, 300 mM NaCl, 500 mM Imidazole, pH 7.4) and dialyzed against Binding buffer. 6xHis-SUMO tag was cleaved with 6×His-ULP1 protease in presence of 1 mM DTT and the tag was removed with Ni-NTA. Ezrin and ezrin FERM were expressed in M15 cells (Qiagen, Hilden, Germany) at 28°C for 4 hr, pellets were lysed in Buffer A

(180 mM $KH_2PO_4$, 180 mM $K_2HPO_4$) and purified as previously described (*Reczek et al., 1997*). Coupling of ezrin FERM to cyanogen bromide (CNBr)-activated Sepharose (Sigma-Aldrich, Saint Louis, MO) was performed as previously described (*Garbett and Bretscher, 2012*). All samples were dialysed against Kinase assay buffer. Non-immobilized proteins were concentrated using Vivaspin 20 MWCO 10 000 (GE Healthcare Bio-Sciences Corp, Pittsburg, PA).

### *In vitro* kinase assay and inhibition assay

Unilamellar liposomes were generated according the following recipes: DOPC (100 mol% DOPC), DOPC:$PIP_2$ (90 mol% DOPC, 10% $PIP_2$), DOPC:DOPS (70 mol% DOPC, 30 mol% DOPS), DOPC: DOPS:$PIP_2$ (80 mol% DOPC, 10 mol% DOPS, 10 mol% $PIP_2$), with 1 mol% DiR near-infrared dye (Avanti Polar Lipids, Alabaster, AL) to aid in visualization. Lipids were mixed, vacuum dried and hydrated in Kinase assay buffer (20 mM Tris pH 7.4, 140 mM NaCl, 1 mM EGTA, 1 mM DTT), followed by extrusion through 100 nm filters to generate liposomes. For micelles, phospholipids were vacuum dried and hydrated in Kinase assay buffer and sonicated in water bath prior to kinase assay. In general, $IP_3$ or phospholipids were added at 90 µM final concentration or at a 1:5 (substrate:phospholipid) molar ratio, and at a maximum concentration of 180 µM due to limited solubility in Kinase assay buffer. Unilamellar liposomes were used at a final concentration of 1 mM resulting in a final concentration of ~100 µM $PIP_2$. Kinase assays were performed in Kinase assay buffer, 500 µM $MgCl_2$ and 200 µM ATP at 37°C for 15 min. 10 nM of purified kinase was used to phosphorylate 18 µM of ezrin, GST-ezrin-CTD or GST-ezin-CTD truncations. In the case of GST-ezrin-CTD+FERM complex, 18 µM GST-ezrin-CTD was pre-incubated with 18 µM FERM for 10 min on ice prior to kinase assay. In the inhibition assay, purified LOK-C was added to individual reaction at concentrations of 0–65 µM. 20% of each kinase reaction was analyzed by SDS-PAGE and immunoblotting.

### Enzyme kinetics

Kinetic assays were initiated by the addition of 500 µM $MgCl_2$ and 200 µM ATP supplemented with 0.1 µL of 3000 Ci/mmol γ-$^{32}$P ATP (PerkinElmer Inc., Waltham, MA) and incubated at 37°C for 15 min. Reactions were stopped by the addition of 12.5 µl of 10% phosphoric acid and processed as previously described (*Hastie et al., 2006*). Briefly, samples were spotted on P81 cellulose paper (GE Healthcare, Pittsburg, PA) and washed three times for 10 min in 75 mM phosphoric acid to remove unincorporated γ-$^{32}$P ATP. Filter papers were allowed to dry and then placed into scintillation vials containing 2.5 mL of Ecoscint fluid (National Diagnostics, Atlanta, GA). $^{32}$P incorporation was measured using PerkinElmer Liquid Scintillation Counter, Tri-Carb 2810TR (PerkinElmer Inc., Waltham, MA).

### Equilibrium binding assay

200 nM of purified LOK-C was diluted in 20 mM Tris pH 7.4, 150 NaCl and incubated with either 0–15 µM GST-ezrin-CTD immobilized on glutathione agarose or 0–18 µM CNBr-FERM for 15 min at RT. Samples were centrifuged at 5000×g for 10 min, supernatants corresponding to 15% of total volume were removed, mixed with sample buffer and loaded on a 10% acrylamide gel for SDS-PAGE. Gels were stained with Coomassie blue and destained in 50% methanol and 10% acetic acid. Densitometric quantification was performed in ImageStudio 5.2.5 (LI-COR Biosciences, Lincoln, NB) by quantifying Coomassie signals after background correction.

### Western blotting and densitometry

Western blots were performed as described (*Hanono et al., 2006*). For analysis of endogenous ezrin phosphorylation, cells were rapidly boiled in 5X reducing sample buffer in kinase buffer, diluted 3-fold, vortexed vigorously, and cleared by centrifugation prior to SDS-PAGE. For densitometric analysis of phospho-ezrin in relation to total ezrin, mouse monoclonal ezrin antibody and rabbit polyclonal pT567 antibody were detected with infrared fluorescent secondary antibodies donkey anti-mouse Alexa Fluor 647 (Thermo Fisher Scientific Cat# A-31571 RRID:AB_162542) or IRDye 800CW goat anti-rabbit (LI-COR Biosciences Cat# 827–08365 RRID:AB_10796098). Membranes were imaged using Odyssey CLx imager (Odyssey CLx , RRID:SCR_014579). Densitometric quantification was performed in ImageStudio 5.2.5 (LI-COR Biosciences, Lincoln, NB) by measuring pT567 and total ezrin after background correction. Total GST-ezrin-CTD was quantified from Coomassie stains. For SLK,

peroxidase conjugated goat anti-rabbit secondary antibody (MP Biochemicals, Santa Ana, CA) was used and developed using ECL western blotting reagent (GE Healthcare, Pittsburg, PA).

## Cell culture and transfection

JEG-3 cells (passage 8–15, STR profiled, mycoplasma negative, ATCC Cat# HTB-36, RRID:CVCL_0363) were maintained in a 5% $CO_2$ humidified atmosphere at 37°C in MEM (Life Technologies, Waltham, MA) with 10% FBS (Life Technologies, Waltham, MA) and 1X GlutaMAX (Life Technologies, Waltham, MA). Phoenix-Ampho retrovirus producer cells (passage 10–20, STR profiled, mycoplasma negative, ATCC Cat# CRL-3213, RRID:CVCL_H716) were maintained in DMEM (Life Technologies, Waltham, MA) with 10% FBS. HEK293 TN cells (passage 10–20, STR profiled, mycoplasma negative, ATCC Cat# CRL-1573, RRID:CVCL_0045) used for Lentivirus production were maintained in DMEM (Life Technologies, Waltham, MA) with 10% FBS and 1 mM sodium pyruvate. Transient transfections used polyethylenimine (PEI) reagent (PolyPlus, Strasbourg, France) by forming DNA-PEI complexes at a ratio of 2.5:1 and 0.3 µg DNA/cm$^2$ of 30–50% confluent JEG-3 cells or 70% confluent Phoenix-Ampho retrovirus producer cells. Transfected cells were processed 24 hr after transfection. Selection was performed with 2 µg/mL puromycin (Sigma-Aldrich, Saint Louis, MO). All siRNAs were transfected at 30 nM into ~25% confluent JEG-3 cells using Lipofectamine RNAiMAX (Invitrogen, Carlsbad, CA) according to the manufacturer's protocol. Cells were processed 48 hr after transfection. Transient transfection in knock-down cells was performed 48 hr after transfection with siRNA and cells were processed after 24 hr.

## Gene deletion using CRISPR/Cas9 system

Deletion of *LOK* gene in JEG-3 cells was achieved using the sgRNA pair: forward 5'-CACCGGTAA-GACTCACCCAGCATGA-3' and reverse 5'-TTTGTCATGCTGGGTGAGTCTTACG-3', targeting exon 3 (*Shalem et al., 2014*). The oligonucleotides were phosphorylated, annealed and cloned into pLentiCRISPR plasmid using a BsmBI restriction site (*Shalem et al., 2014*). Virus vectors were produced in HEK293 TN cells by co-transfection with psPAX2 (a gift from Didier Trono [Addgene plasmid # 12260]) and pCMV-VSV-G (*Stewart et al., 2003*) using Lipofectamine 2000 (Thermo Fisher Scientific Inc., Waltham, MA). Collected viral vectors were used to infect JEG-3 cells in presence of 8 µg/mL polybrene. JEG-3 cells were maintained under 2 µg/mL puromycin selection and subjected to three rounds of recloning for isolation of LOK knockout monoclones.

## Immunofluorescence and image analysis

JEG-3 cells seeded on coverslips were fixed in 3.7% formaldehyde/PBS for 15 min at RT, washed with PBS and permeabilized with 0.1% Triton X-100/PBS for 5 min, washed, and blocked with 5% FBS/PBS for 20 min at RT. Cells were stained for 1 hr at RT with primary antibody in 5% FBS/PBS. Secondary antibody and Alexa Fluor-conjugated phalloidin staining was performed for 1 hr at RT. Cells were washed with PBS and mounted using Vectashield reagent (Vector Laboratories Cat# H-1000 RRID:AB_2336789) and imaged using a CSU-X spinning disk microscope (Intelligent Imaging Innovations, Santa Monica, CA) with spherical aberration correction device, 100×/1.46 numerical aperture (NA) objective on an inverted microscope (DMI6000B), and an HQ2 charge-coupled device (CCD) camera (Photometrics, Tucson, AZ) using SlideBook 6 (Intelligent Imaging Innovations, Denver, CO). Maximum intensity projections were assembled in SlideBook 6.

## Scoring the apical surface phenotype of JEG-3 cells

The presence of microvilli on apical surfaces of JEG-3 cells was scored as described previously (*Hanono et al., 2006*). Briefly, JEG-3 cells were stained for ezrin, F-actin and the expressed GFP fusion protein to indicate the apical membrane phenotype. Twenty cells were counted for each experiment and scored as having normal microvilli or no microvilli. Kinase-suppressed JEG-3 cells were stained for ezrin and F-actin. Twenty cells were counted and scored for each condition as described above.

## Statistics

Michaelis-Menten kinetics were fitted in GraphPad Prism (Graphpad Prism, RRID:SCR_002798) using the following equation,

$$V = (Vmax \cdot [S])/(Km + [S]) \tag{1}$$

The turnover number, $k_{cat}$, was calculated using the following equation,

$$V = (Et \cdot kcat \cdot [S])/(Km + [S]) \tag{2}$$

For calculation of $IC_{50}$, data were normalized to LOK/ezrin+$PIP_2$ or LOK/ezrin-CTD data sets in which phosphorylation signal at 0 µM LOK-C, defined as 'Top', was set to 100%. Data were fitted in GraphPad Prism using the four parameter logistic equation,

$$pT567/Total = Bottom + (Top - Bottom)/\left(1 + 10^{(LogIC50 - [I]) \cdot HillSlope}\right) \tag{3}$$

Data are represented as means ± standard error (S.E.) from three independent experiments. Two-way ANOVA followed by Tukey's multiple comparisons test was performed using GraphPad Prism.

## Acknowledgements

We are grateful to Dr. Yuxin Mao for critically reading this manuscript.

## Additional information

### Funding

| Funder | Grant reference number | Author |
| --- | --- | --- |
| Wenner-Gren Foundation | | Thaher Pelaseyed |
| National Institutes of Health | GM036552 | Thaher Pelaseyed<br>Raghuvir Viswanatha<br>Cécile Sauvanet<br>Anthony Bretscher |
| National Institutes of Health | GM048430 | Joshua J Filter<br>Michael L Goldberg |
| Swedish Cystic Fibrosis Foundation | | Thaher Pelaseyed |
| Birgit and Hellmuth Hertz Foundation | | Thaher Pelaseyed |

The funders had no role in study design, data collection and interpretation, or the decision to submit the work for publication.

### Author contributions

TP, Conceptualization, Designed all experiments and carried out all aspects of experiments and collected the data (except otherwise stated), Data curation, Formal analysis, Validation, Investigation, Visualization, Methodology, Writing—original draft, Writing—review and editing; RV, Assisted with image acquisition and data analysis, Contributed to experimental design, Assisted with preparing the revised manuscript; CS, Assisted conceptualization and methodology, Contributed to experimental design, Assisted with preparing the manuscript; JJF, Carried out enzyme kinetics experiments, Assisted with data analysis, Contributed to experimental design, Assisted with preparing the manuscript; MLG, Assisted with conceptualization and methodology for enzyme kinetics, Contributed to experimental design, Assisted with preparing the manuscript; AB, Conceptualization, Resources, Formal analysis, Supervision, Funding acquisition, Investigation, Writing—original draft, Writing—review and editing

### Author ORCIDs

Thaher Pelaseyed, http://orcid.org/0000-0002-6434-3913
Michael L Goldberg, http://orcid.org/0000-0003-0200-0277
Anthony Bretscher, http://orcid.org/0000-0002-1122-8970

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
