## [Decision Letter]

Thank you for submitting your article "Ezrin activation by LOK phosphorylation involves a PIP2-dependent wedge mechanism" for consideration by *eLife*. Your article has been favorably evaluated by Jonathan Cooper (Senior Editor) and three reviewers, one of whom, Pekka Lappalainen (Reviewer #1), is a member of our Board of Reviewing Editors.

The reviewers have discussed the reviews with one another and the Reviewing Editor has drafted this decision to help you prepare a revised submission.

Summary:

Ezrin/radixin/moesin (ERM) family proteins function as linkers between the cortical actin cytoskeleton and plasma membrane. They exist in an autoinhibited conformation that is released by interaction of the N-terminal FERM domain with plasma membrane phosphoinositides, and subsequent phosphorylation at a threonine residue, which in an autoinhibited conformation is buried between the FERM domain and the C-terminal actin-binding domain. Here, Pelaseyed et al. examined the mechanisms by which phosphoinositide-interactions control ezrin phosphorylation by the LOK kinase. They demonstrate that the C-terminal domain of LOK serves as a wedge that binds between the FERM domain and the C-terminal actin-binding domain in a PIP2-primed ezrin to allow the phosphorylation of ezrin by the N-terminal kinase domain of LOK. Furthermore, the authors provide evidence that the central a-helical domain of ezrin functions as a 'spring', which helps to loosen binding between the C-terminal and FERM domains upon interaction with PIP2.

All three reviewers stated that this is a very important piece of work as it provides, for the first time, solid evidence that ezrin phosphorylation takes place only if it is bound to PIP2, and elucidates the underlying molecular mechanism. This has important implications for our understanding on how ezrin is activated at particular sites of the cell, including microvilli. However, a few additional experiments are required to strengthen the main conclusions presented and to confirm the *in vivo* relevance of the findings.

Essential revisions:

1) The majority of the in vitro experiments (with the exception of the ones presented in Figure 1) were apparently performed with PIP2-micelles, which display very different lipid composition and geometry compared to the plasma membrane. Thus, at least the most critical experiments (e.g. the ones presented in Figure 1 and Figure 2) should be repeated by using unilamellar vesicles with a more physiological lipid composition. The authors should also discuss, how important the micellar/proper bilayer structure of PIP2 is for ezrin activation, because soluble IP_3_ does not apparently activate ezrin?

2) The role of central a-helical region as a 'spring' is interesting. Is there any evidence in the literature to support this notion? To confirm the *in vivo* relevance of this interesting observation, the authors should express ezrin(del)a1 and ezrin(del)a7 proteins (used in Figure 2) in cells, and test whether they fail to undergo PIP2-dependent phosphorylation and whether they still properly localize to the plasma membrane/microvilli.

3) One missing piece of information related to the model (presented in Figure 6) is how LOK itself is localized to the plasma membrane. The same laboratory reported that this localization is governed by the C-terminus of LOK (PMID: 23209304). It is therefore very tempting to postulate that ezrin could itself regulate the localization of its own kinase by binding to the C-terminal domain of LOK. Did the authors test this hypothesis in cells depleted for ERM proteins?

---

## [Author Response]

Essential revisions:

1) The majority of the in vitro experiments (with the exception of the ones presented in Figure 1) were apparently performed with PIP2-micelles, which display very different lipid composition and geometry compared to the plasma membrane. Thus, at least the most critical experiments (e.g. the ones presented in Figure 1 and Figure 2) should be repeated by using unilamellar vesicles with a more physiological lipid composition. The authors should also discuss, how important the micellar/proper bilayer structure of PIP2 is for ezrin activation, because soluble IP_3_ does not apparently activate ezrin?

Some experiments were performed with PIP_2_ micelles and it was suggested that Figure 1, and Figure 2 should be repeated using unilamellar vesicles with/without PIP_2_. We have now done these experiments for Figure 1 and Figure 2, with the same outcome as with PIP_2_ alone (Figure 1 did not involve PIP_2_, so it is not clear why it was listed). Please see Figure 1—figure supplement 1 and Figure 2—figure supplement 1. We have also added a discussion regarding why PIP_2_ in micelles or unilamellar liposomes primes ezrin whereas IP_3_ does not. Please see “Discussion”, second paragraph.

2) The role of central a-helical region as a 'spring' is interesting. Is there any evidence in the literature to support this notion? To confirm the in vivo relevance of this interesting observation, the authors should express ezrin(del)a1 and ezrin(del)a7 proteins (used in Figure 2) in cells, and test whether they fail to undergo PIP2-dependent phosphorylation and whether they still properly localize to the plasma membrane/microvilli.

Additional experiments were requested relating to two mutants in which heptad repeats were removed from the anti-parallel coiled-coil region of ezrin. During subcloning experiments to express the two heptad mutants in cultured cells, we uncovered a previously overlooked mutation at the N-terminal end of the protein. When this mutation was removed in the context of the two heptad mutants, the resulting variants could be phosphorylated *in vitro* by LOK in a PIP_2_-dependent manner using both PIP_2_ micelles and unilamellar vesicles (Figure 7). Moreover, in cells depleted of all endogenous ERM proteins using CRISPR, the heptad mutants could be phosphorylated and restored microvilli (Figure 8) – thus they appear functional. Therefore, the reported inability of the heptad mutants to serve as a substrate for LOK is not due to the heptad mutations, but rather due to the N-terminal mutation. While this is interesting, it does not support our speculation that the helical region acts like a spring. The important data that the helical region transmits a PIP_2_-dependent conformational signal from the N-terminal FERM domain to prime the ezrin-CTD for phosphorylation is accurate. We have adjusted Figure 2 and the text to remove mention of the heptad mutants, and the appropriate speculation from the discussion. Although we are relieved to have caught this error before publication, it does not change the outcome of this study and the model presented in Figure 6 is still valid. As Dr. Cécile Sauvanet uncovered the error and provided some of this data, she has been added as an author.

Author response image 1.Full length LOK can phosphorylate ezrin, ezrin ∆α1 and ezrin ∆α7 in a PIP2-dependent manner.Full length LOK (10nM) was incubated with 18µM ezrin or ezrin variant and ATP for 10 min at 37°. Samples were resolved by SDS-PAGE and blotted for ezrin (red) and pT567 (green).**DOI:**
http://dx.doi.org/10.7554/eLife.22759.018

Author response image 2.(**A**) I*n vivo* kinase assay showing that Ezrin WT, Ezrin Δα1 and Ezrin Δα7 are phosphorylated in Ezrin/Radixin deleted cells. Cells were transfected with pQCXIP empty (Empty vector), pQCXIP-Ezrin WT, pQCXIP-Ezrin Δα1 or pQCXIP- Ezrin Δα7. After 24 hr, cells were lysed in 2X sample buffer. Samples are then resolved by SDS-PAGE and analyzed by immunoblotting. Total ezrin is showed in red and phosphorylation of T567 in green in dual color western blots. (**B**) Maximum projections of Ezrin/Radixin deleted cells showing that Ezrin WT, Ezrin Δα1 and Ezrin Δα7 can rescue microvilli. Cells were transfected with pQCXIP empty (vector control), pQCXIP-Ezrin WT, pQCXIP-Ezrin Δα1 or pQCXIP- Ezrin Δα7. After 24hrs, cells are fixed then stained using phalloidin 660 and antibodies against EBP50 and Ezrin.**DOI:**
http://dx.doi.org/10.7554/eLife.22759.019

3) One missing piece of information related to the model (presented in Figure 6) is how LOK itself is localized to the plasma membrane. The same laboratory reported that this localization is governed by the C-terminus of LOK (PMID: 23209304). It is therefore very tempting to postulate that ezrin could itself regulate the localization of its own kinase by binding to the C-terminal domain of LOK. Did the authors test this hypothesis in cells depleted for ERM proteins?

The mechanism by which LOK is targeted to the apical membrane is currently under study. The possibility that it involves binding to activated ezrin in a positive feed-back loop is a very attractive possibility, but so far we have no clear data supporting or refuting this model. We do know that in cells depleted of ERM proteins, LOK is delocalized, but since those cells lack microvilli, it is not possible to conclude that LOK simply binds activated ezrin.